# Early postmortem mapping of SARS-CoV-2 RNA in patients with COVID-19 and the correlation with tissue damage

Stefanie Deinhardt-Emmer[1†]*, Daniel Wittschieber[2†], Juliane Sanft[2], Sandra Kleemann[2], Stefan Elschner[2], Karoline Frieda Haupt[1], Vanessa Vau[1], Clio Häring[3], Jürgen Rödel[1], Andreas Henke[3], Christina Ehrhardt[3], Michael Bauer[4], Mike Philipp[5], Nikolaus Gaßler[6], Sandor Nietzsche[7], Bettina Löffler[1‡], Gita Mall[2‡]

[1]Institute of Medical Microbiology, Jena University Hospital, Jena, Germany; [2]Institute of Forensic Medicine, Jena University Hospital, Jena, Germany; [3]Section of Experimental Virology, Institute of Medical Microbiology, Jena University Hospital, Jena, Germany; [4]Department of Anaesthesiology and Intensive Care Medicine, Jena University Hospital, Jena, Germany; [5]Department of Anaesthesiology and Intensive Care Medicine, Greiz General Hospital, Greiz, Germany; [6]Section of Surgical Pathology, Institute of Forensic Medicine, Jena University Hospital, Jena, Germany; [7]Department of Electron Microscopy, Jena University Hospital, Jena, Germany

*For correspondence:
stefanie.deinhardt-emmer@med.uni-jena.de

[†]These authors contributed equally to this work
[‡]These authors also contributed equally to this work

Competing interests: The authors declare that no competing interests exist.

**Abstract** Clinical observations indicate that COVID-19 is a systemic disease. An investigation of the viral distribution within the human body and its correlation with tissue damage can aid in understanding the pathophysiology of SARS-CoV-2 infection. We present a detailed mapping of the viral RNA in 61 tissues and organs of 11 deceased patients with COVID-19. The autopsies were performed within the early postmortem interval (between 1.5 and 15 hr, mean: 5.6 hr) to minimize the bias due to viral RNA and tissue degradation. Very high viral loads ($>10^4$ copies/ml) were detected in most patients' lungs, and the presence of intact viral particles in the lung tissue could be verified by transmission electron microscopy. Interestingly, viral RNA was detected throughout various extrapulmonary tissues and organs without visible tissue damage. The dissemination of SARS-CoV-2-RNA throughout the body supports the hypothesis that there is a maladaptive host response with viremia and multiorgan dysfunction.

## Introduction

In December 2019, several cases of pneumonia caused by a novel *Betacoronavirus* called SARS-CoV-2 were first described in the city of Wuhan in China (*Zhu et al., 2020*); the disease was thereafter named 'coronavirus disease 2019' (COVID-19) (*Lu et al., 2020*). Within a few months, the initially localized outbreak spread to countries all over the globe and was declared a pandemic (*El Zowalaty and Järhult, 2020*). At present, more than 121 million SARS-CoV-2 infections have been reported (*Dong et al., 2020*). The number of deaths attributed to COVID-19 has exceeded 2.6 million worldwide (*Dong et al., 2020*).

COVID-19 occurs with varying degrees of severity. While approximately 81% of COVID-19 patients experience mild symptoms, 14% suffer from respiratory distress, and 5% have to be hospitalized (*Wu and McGoogan, 2020*; *Wiersinga et al., 2020*). Of these, 20% enter a critical condition with respiratory failure, endovascular complications, or multiple organ dysfunction.

**eLife digest** Since the discovery of the new coronavirus that causes COVID-19, scientists have been scrambling to understand the different features of the virus. While a lot more is now known about SARS-CoV-2, several key questions have proved more difficult to answer. For example, it remained unclear where the virus travels to in the body and causes the most harm.

To help answer this question, Deinhardt-Emmer, Wittschieber et al. performed postmortem examinations on 11 patients who had recently died of COVID-19. After sampling 61 different organs and tissues from each patient, several tests were used to detect traces of SARS-CoV-2. The experiments showed that the largest pool of SARS-CoV-2 was present in the lungs, where it had caused severe damage to the alveolae, the delicate air sacs at the end of the lungs' main air tubes.

Small amounts of the virus were also detected in other organs and tissues, but no severe tissue damage was seen. In addition, Deinhardt-Emmer, Wittschieber et al. found that each patient had increased levels of some of the proteins involved in inflammation and blood clotting circulating their bloodstream. This suggests that the inflammation caused by SARS-CoV-2 leads to an excessive immune reaction throughout the entire body.

This research provides important new insights into which areas of the body are most impacted by SARS-CoV-2. These findings may help to design more effective drug treatments that target the places SARS-CoV-2 is most likely to accumulate and help patients fight off the infection at these regions.

Gastroenterological and neurological symptoms have been reported in 36.4% and 18.6% of COVID-19 patients, respectively, in case studies (*Wiersinga et al., 2020*; *Pan et al., 2020*; *Mao et al., 2020*). The clinical observations suggest that COVID-19 is a systemic disease.

While little information has been available to date about the molecular regulation of SARS-CoV-2 infections, angiotensin-converting enzyme 2 (ACE2) and transmembrane protease serine 2 (TMPRSS2), two membrane-bound proteins, have been shown to be crucial for the entry of the virus into cells (*Wang et al., 2020*; *Hoffmann et al., 2020*). ACE2 is expressed not only in the epithelia of the lung but also in several other epithelial, endothelial, heart, and renal tissues (*Hamming et al., 2004*). SARS-CoV-2 viral replication and pathogenesis in organisms are currently not well understood due to the lack of appropriate models (*Bar-On et al., 2020*). One crucial step to elucidate viral pathogenesis is the investigation of the distribution of the virus within the entire body.

In the present study, we (1) included full autopsies, (2) performed autopsies in the early postmortem interval (1.5–15 hr, mean: 5.6 hr), (3) dissected organs and tissues without prior fixation in formalin, (4) measured SARS-CoV-2 RNA in a high number of samples, (5) correlated the viral load with tissue damage using comprehensive histopathological investigations, (6) visualized virus particles in pulmonal tissue samples by means of transmission electron microscopy (TEM), and (7) determined the postmortem serum levels of inflammatory cytokines and prothrombotic factors. Sampling was performed in the very early postmortem interval to provide reliable viral RNA (vRNA) measurements and enabled us to obtain blood serum and well-preserved tissue samples for ultrastructural analysis.

## Results

### SARS-CoV-2 vRNA is detectable in various organs and tissues

The investigation of COVID-19 patients included a full characterization of the clinical characteristics and parameters (*Table 1*). In detail, patients 1–7 received intensive care; patients 1–6 were mechanically ventilated, whereas patient 7 was subjected to ECMO (extracorporeal membrane oxygenation). Patients 1–2 and 5–6 were treated with lopinavir/ritonavir. Patients 8–10 were not subjected to intensive care or ventilation according to their patient provisions.

Our results show that patients 1–10 died of COVID-19, whereas patient 11 suffered from a metastasized squamous cell carcinoma of the cervix and died of an ileus following peritoneal carcinosis (*Table 2* provides an overview of the macro- and micromorphological autopsy findings). Patient 11 had contracted COVID-19 and received intensive care treatment but was not ventilated. Interestingly, autopsy detected previously undiagnosed malignancies in patients 2 (chronic lymphatic

**Table 1.** Clinical characteristics of deceased COVID-19 patients.

| | Patient 1 | Patient 2 | Patient 3 | Patient 4 | Patient 5 | Patient 6 | Patient 7 | Patient 8 | Patient 9 | Patient 10 | Patient 11 |
|---|---|---|---|---|---|---|---|---|---|---|---|
| Sex | m | m | m | m | m | m | m | f | f | f | f |
| Age [years] | 82 | 66 | 78 | 54 | 80 | 64 | 64 | 87 | 83 | 85 | 52 |
| BMI [kg/m$^2$] | 23.8 | 31.5 | 25.2 | 28.3 | 28.8 | 35.4 | 24.6 | 24.6 | 28.9 | 26.2 | 21.4 |
| Pre-existing medical conditions | AF, DM, autoimmune pancreatitis, purpura pigmentosa | aHT | aHT, DM, CRF, PAD, urosepsis shortly before COVID-19 | None known | DM, CRF, CHF | GPA, CRF, COPD, aHT, AF, DM | AS, COPD | CHF, CRF, DM, past stroke epilepsy, erysipelas shortly before COVID-19 | aHT, DM, CRF, 1-vessel-CHD, AF, PAD | CHF, CRF, DM | Cervical carcinoma |
| Hospitalization [d] | 7 | 5 | 30 | 10 | 9 | 20 | 15 | 12 | 7 | 9 | 4 |
| ICU [d] | 7 | 5 | 7 | 8 | 8 | 17 | 9 | 0 | 0 | 0 | 4 |
| Mechanical ventilation [d] | 7 | 4 | 7 | 7 | 8 | 16 | 9 (ECMO) | 0 | 0 | 0 | 0 |
| Antiviral drugs | Lopinavir, ritonavir | Lopinavir, ritonavir | None | None | None | Lopinavir, ritonavir | Lopinavir, ritonavir | None | None | None | None |
| Cause of death (acc. to clinic) | MOF | Pulmonary embolism | MOF | Lung failure | Suspected myocardial infarction | Lung failure | Lung failure | Respiratory failure | Pneumonia | Pneumonia | Ileus |

aHT – arterial hypertension, AF – atrial fibrillation, AS – atherosclerosis, CHF – chronic heart failure, CHD – coronary heart disease, COPD – chronic obstructive pulmonary disease, CRF – chronic renal failure, DM – diabetes mellitus, ECMO - Extracorporeal membrane oxygenation, f – female, GPA – granulomatosis with polyangiitis (Wegener's Granulomatosis), ICU – intensive care unit, MOF – multiple organ failure, PAD – peripheral artery disease.

leukemia, CLL) and 10 (endometrial carcinoma). In addition, patient 6 had an incidentaloma of the thyroid gland.

Amplification of the E-gene of SARS-CoV-2 by using qRT-PCR detected a very high to high mean viral load in the lungs of all patients (*Figure 1*). Patients 1–10 showed high viral loads, which were as high as $10^7$ RNA copies/ml (*Figure 1—figure supplement 1*). High to moderate to low viral loads were detected in other structures of the respiratory tract, such as the mesopharynx, epiglottis and trachea, in patients 1–8. In patient 11, with a non-COVID-19-associated cause of death, vRNA could only be detected in moderate to low amounts in the trachea.

Patients 1–10 also showed variable (very high to very low) viral loads in at least two samples obtained from the lymphatic system. Lymphatic structures with topological relationships to the respiratory tract were always positive for vRNA.

Of the patients subjected to intensive care treatment (patients 1–7), patients 1, 3, 4, and 5 exhibited moderate to very low viral loads in the cardiac samples (*Figure 1—figure supplement 2*) Patients 2, 6, and 7 exhibited no vRNA in the heart muscle. The vascular samples exhibited higher viral loads overall than the cardiac samples in the majority of patients.

Viral RNA was detected in the blood only for patients 3–5. Patients 1, 2, 6, and 7, who were treated with lopinavir/ritonavir, were negative for vRNA in blood. Viral RNA was present in the bone marrow of all three patients who tested positive for vRNA in blood and was also found in an additional three patients who tested negative for vRNA in blood.

Patients 3–5 had vRNA in variable amounts throughout the small and large intestine. Patients 6 and 7 tested negative for all 12 gastrointestinal samples. Of note, in patient 9, almost all the gastrointestinal samples exhibited moderate to very high viral loads.

Viral RNA could also be detected in endocrine organs, in the urinary tract and in the reproductive organs. Only patients 3–5 tested positive for vRNA in the central nervous system. Skin (abdominal), subcutaneous tissue (abdominal) and skeletal muscle (psoas major) tested negative in all patients.

Due to the very early postmortem interval in which the autopsies were performed, we were able to verify the vRNA findings via TEM of a lung sample from patient 3 by detecting intact SARS-CoV-2 viral particles within lung fibrocytes (*Figure 2*).

**Table 2.** Autopsy findings.

| | | Patient 1 | Patient 2 | Patient 3 | Patient 4 | Patient 5 | Patient 6 | Patient 7 | Patient 8 | Patient 9 | Patient 10 | Patient 11 |
|---|---|---|---|---|---|---|---|---|---|---|---|---|
| PMI [h] | | 3.5 | 2.25 | 7.5 | 9.5 | 15.0 | 2.33 | 1.5 | 5.8 | 5.0 | 6.0 | 3.5 |
| Lung weight | | R 1550 g L 1240 g | R 940 g L 760 g | R 1170 g L 790 g | R 1860 g L 1640 g | R 1370 g L 990 g | R 970 g L 570 g | R 890 g L 770 g | R 610 g L 480 g | R 550 g L 680 g | R 530 g L 400 g | R 760 g L 590 g |
| Lung macro* | Edema | +++ | ++ | ++ | ++ | ++ | + | + | + | + | ++ | + |
| | Hyperemia | +++ | + | +++ | ++ | ++ | + | + | + | + | ++ | + |
| | Hemorrhage | +++ | + | +++ | ++ | ++ | ++ | + | - | + | ++ | + |
| | Texture | reduced | reduced | reduced | reduced | reduced | enhanced | enhanced | enhanced | reduced | reduced | reduced |
| | Infarction | - | - | + | + | - | - | - | - | - | - | - |
| Lung micro* | DAD, exsudative phase | +++ | ++ | ++ | ++ | +++ | + | + | + | + | ++ | + |
| | DAD, proliferative phase | ++ | - | - | + | + | +++ | +++ | +++ | ++ | ++ | - |
| | Multinucleated giant cells | +++ | ++ | ++ | ++ | ++ | + | + | - | ++ | + | + |
| | Squamous metaplasia | +++ | + | ++ | ++ | ++ | + | ++ | ++ | ++ | +++ | + |
| | Megakaryocytes | - | - | - | - | - | + | + | ++ | + | - | + |
| | Lymphocytic infiltrates | + | + | - | - | ++ | - | +++ | - | - | + | - |
| | Vasculitis | - | + | + | + | - | - | - | + | + | - | - |
| | Stasis / fibrin thrombi | + | + | - | - | - | - | - | + | + | + | tumor |
| | Emboli | - | ++ | - | + | + | - | - | - | - | + | tumor |
| | Superinfection | - | - | Fungal | - | - | bacterial | - | - | bacterial | bacterial | - |
| Heart weight | | 480 g | 600 g | 550 g | 430 g | 810 g | 610 g | 480 g | 380 g | 360 g | 300 g | 350 g |
| Heart macro | | Pericarditis, 2-v-CHD | conc. HT | 3-v-CHD | Unremarkable | exc. HT | exc. HT | - | 2-v-CHD | 3-v-CHD | 1-v-CHD | un- remarkable |
| Geart micro | | Moderate fibrosis | Moderate fibrosis | Chronic ischemia | Unremarkable | amyloidosis | slight fibrosis | slight fibrosis | slight fibrosis | moderate fibrosis | atrophy, fibrosis | invasive metastases |
| Further autopsy findings (besides age-related or pre-existing) | | Steatosis hepatis | NASH | - | CLL, ICH, spleen infarction | severe cardiac amyloidosis | hepatic siderosis | - | adrenal thrombosis | - | endometrial carcinoma (apT1a) | cervical carcinoma (apT4) |
| Cause of death (acc. to autopsy)† | | COVID-19 (HP) | COVID-19 (pulmonary embolism) | COVID-19 (HP) | COVID-19 (HP) | COVID-19 (HP) | COVID-19 (CCP) | COVID-19 (CCP) | COVID-19 (CCP) | COVID-19 (BP+HP) | COVID-19 (BP+HP) | malignant tumor disease |

BP – bronchopneumonia, CCP – chronic carnifying pneumonia, CHD – coronary heart disease, CLL – chronic lymphatic leukemia, DAD – diffuse alveolar damage, HP – hemorrhagic pneumonia, HT – hypertrophy, conc. = concentric, exc. = excentric, ICH – intracerebral hemorrhage, NASH – non-alcoholic steatosis hepatitis, PMI – postmortem interval (time between death and autopsy), v – vessel.

*Semi-quantitative evaluation: no (-), few (+), moderate (++), very much (+++).

†Supplemented term within brackets describes the dominant finding that caused death by COVID-19.

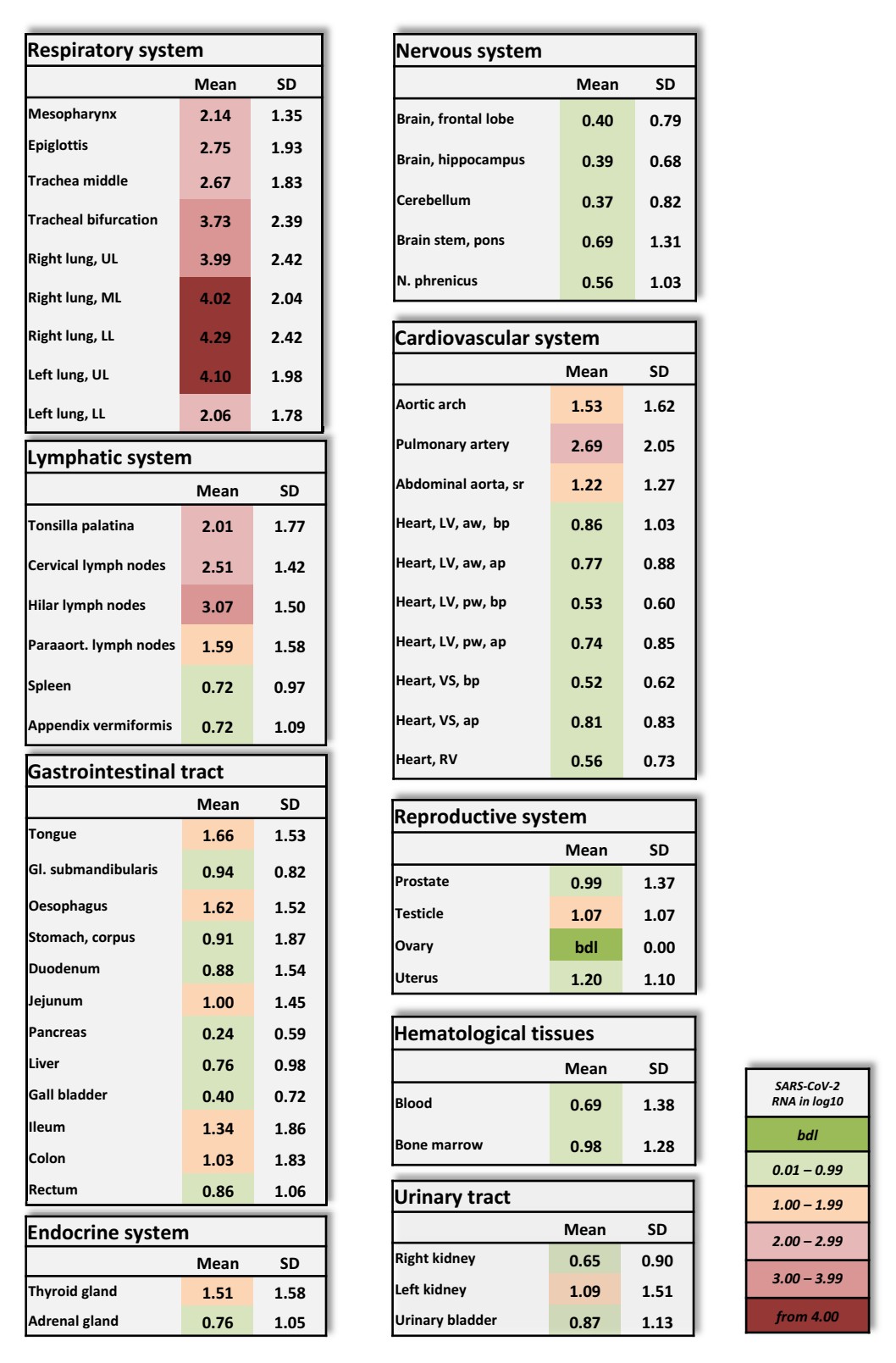

**Figure 1.** Overview of SARS-CoV-2 vRNA throughout the human body. Postmortem determination of SARS-CoV-2 RNA with qRT-PCR of homogenized organs and tissues in copies/ml represented as decadic logarithm of 11 patients with mean value and standard deviation (SD) of the following systems: respiratory system, lymphatic

*Figure 1 continued on next page*

*Figure 1 continued*

system, gastrointestinal tract, urinary tract, nervous system, cardiovascular system, hematological tissues, reproductive system, and endocrine system. Intensity of colors describes the amount of vRNA. Abbrev.: bdl (below detection limit), UL (upper lobe), ML (middle lobe), LL (lower lobe), LV (left ventricle), sr (suprarenal), VS (ventricular septum), RV (right ventricle), aw (anterior wall), pw (posterior wall), bp (basal part), ap (apical part), paraaort. (paraaortal).

The online version of this article includes the following source data and figure supplement(s) for figure 1:

**Source data 1.** Postmortem determination of SARS-CoV-2.

**Figure supplement 1.** Individual vRNA load of the respiratory system, the lymphatic system, the cardiovascular system, and hematological tissues.

**Figure supplement 2.** Individual vRNA load of the gastrointestinal tract, the endocrine system, the urinary tract, the nervous system, and the reproductive system.

## Proinflammatory and prothrombotic parameters

To analyze the proinflammatory responses of the 11 patients, we measured interleukin (IL)−6 (*Figure 3a*) and IL-8 (*Figure 3b*) postmortem. Both parameters showed significantly elevated serum levels in all cases compared to the levels in five healthy volunteers, who served as controls.

Since abnormalities in the coagulation system were described in COVID-19 patients (*Connors and Levy, 2020*), we analyzed the prothrombotic parameters of the blood of the deceased patients. Disseminated intravascular coagulation was described previously, and one study suggested that the role of coagulation is to limit infection dissemination (*Antoniak, 2018*). Additionally, coagulation processes cause hyperinflammatory responses in viral infection (*Yang and Tang, 2016*).

Our results showed significantly higher serum levels of tissue plasminogen activator (tPa) (*Figure 3c*) in COVID-19 patients than in healthy volunteers. P-Selectin (*Figure 3d*), a cell adhesion molecule in platelets necessary for the recruitment of platelets and binding to the endothelium (*Furie and Furie, 2004*), was found in all patients in serum at levels significantly higher than those in controls. D dimer serum levels (*Figure 3e*) were slightly elevated in patients. However, no significant

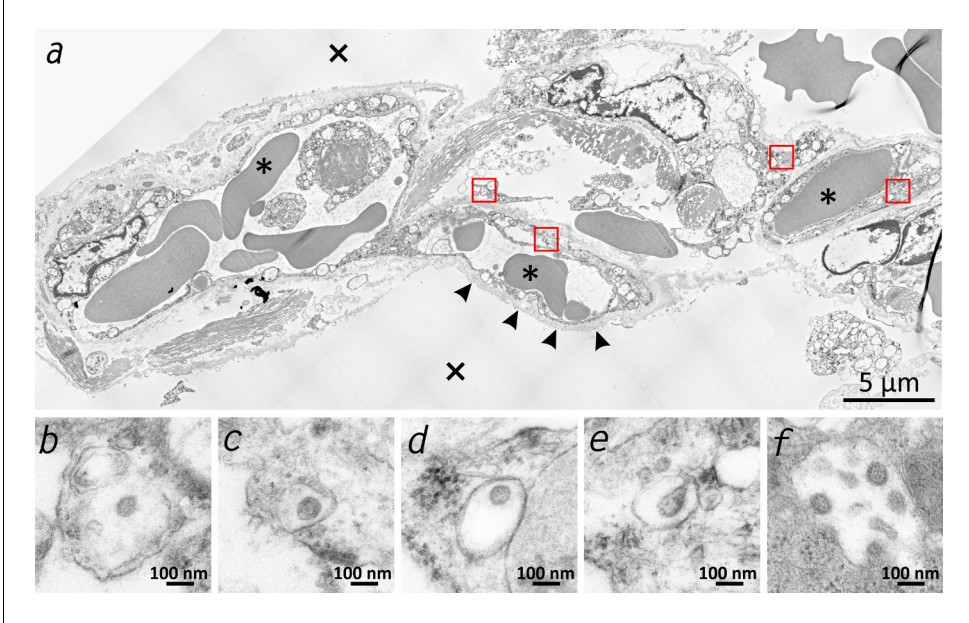

**Figure 2.** Transmission electron microscopic image of the lung tissue of patient 3. (**a**) Alveolar septum showing intact capillaries with erythrocytes (asterisk) and the air space (cross). The blood-air barrier is damaged as the pneumocytes are missing and the basal membrane is exposed to air (arrowheads). (**b–e**) Close-ups of the four boxed regions in (**a**) from left to right showing SARS-CoV-2 virus particles encased in plasmatic vesicles of alveolar fibrocytes. (**f**) Reference image of SARS-CoV-2 virus particles proliferated in cell culture (Vero76).

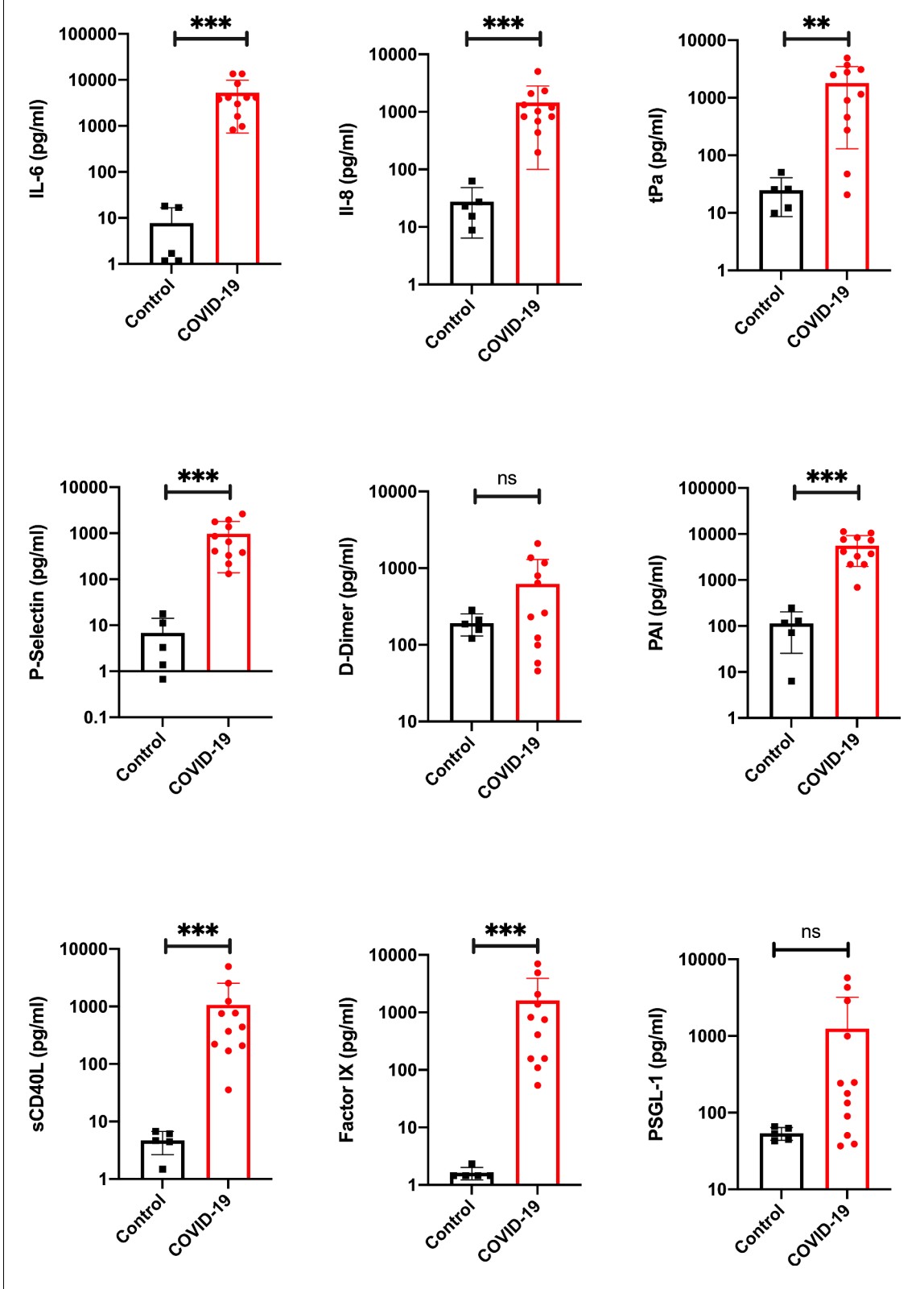

**Figure 3.** Proinflammatory and prothrombotic factors. Blood analysis of patients 1–11 by using Legendplex Panel (Biolegend, CA, USA) of the proinflammatory cytokines Interleukin (IL)—6 (a) and IL-8 (b) as well as tissue plasminogen activator (tPa) (c), P-Selectin (d), D-Dimer (e), Plasminogen activator inhibitor-1 (PAI) (f), soluble (s) CD40ligand(L) (g), Factor IX (h), and the P-selectin glycoprotein ligand 1 (PSGL-1) (i) in pg/ml compared to the mean of five controls (Control, healthy volunteers). Unpaired t-test, Mann-Whitney test p<0.005 ***; p<0.05 **; ns=not significant.

*Figure 3 continued on next page*

*Figure 3 continued*

The online version of this article includes the following source data for figure 3:

**Source data 1.** Blood analysis of patients 1–11 by using the Legendplex Panel (BioLegend, san Diego, CA, USA) .

elevation could be found. The level of plasminogen activator inhibitor (PAI, *Figure 3f*), an important inhibitor of tPa (*Cesari et al., 2010*), was significantly elevated in the blood serum of all patients.

The level of a biomarker with potential for cardiovascular risk stratification, sCD40L (*Figure 3g*), was significantly elevated in all patients. Furthermore, coagulation factor IX (*Figure 3h*) was measured at significantly elevated levels in the blood serum of all patients; it has been proposed to function as a mediator between viruses and cells (*Lenman et al., 2011*). The level of P-Selectin Glykoprotein Ligand-1 (PSGL-1) was determined to be not significantly higher in patients than in the healthy volunteer group.

## Macro- and micromorphologic findings

Macroscopic signs of severe and extensive lung damage were found in all patients. In patients 1, 3–5, 9, and 10, the lung tissue was full of blood (hyperemia) and fluid (edema) and showed weakened consistency (*Figure 4a,b*). In contrast, the lungs of patients 6–8 displayed a firmer and more consolidated pattern with only a low degree of edema and hyperemia (*Figure 4c–e*). In the lungs of patient 3, nodular demarcated damage was found that was correlated with fungal superinfection (*Figure 4f*). Patients 6, 9, and 10 had purulent bronchitis and bronchopneumonia due to bacterial superinfection, and patient nine exhibited severe pharyngitis.

Several features of coagulopathies were found, including infarction of the lung (*Figure 4g*) and the spleen (*Figure 4h*) as well as fulminant thromboses of the periprostatic venous plexus (*Figure 4i*) and hemorrhage of the cerebellum (*Figure 4j*). Vascular stasis and fibrinous thrombi were present in patients 1–2 and 8–10. Thrombemboli were found in patients 2, 4, 5, and 10 to a variable extent. In patient 2, pulmonary embolisms were fatal.

Microscopically, the lung tissues from patients 1, 3–5, 9, and 10 revealed changes consistent with the early (exudative) phase of diffuse alveolar damage (DAD). The consistent acute changes included severe intraalveolar and interstitial hemorrhages (*Figure 5a*, *Figure 5b*), architectural injuries with a diffuse alveolar damage pattern (involving hyaline membranes, fibrinous edema and interstitial proliferation), sporadic signs of cellular inflammation (mostly involving lymphocytes and a few plasma cells) and severe loss of structured pneumocytes. Frequently, cells with an enlarged cytoplasm and large nuclei were found to be admixed with multinucleated giant cells and to show features of squamous metaplasia and a pattern of bronchiolization (*Figure 5a,b,c*). Enlarged alveolar cells were detached from the alveolar wall (*Figure 5d*). The clusters of enlarged cells were strongly positive for AE1/3 (*Figure 5e*), but only a few cells were colabeled with TTF1 (*Figure 5f*). The lung histology in patients 6–8 displayed a pattern similar to the latter (proliferative) phase of diffuse alveolar damage (DAD). Giant cells and cell aggregates resembling squamous metaplasia were frequently found and sometimes accompanied by fibroblastic proliferation (*Figure 5g,h*).

While the upper lobes of the lung of patient 2 showed only moderate emphysema (*Figure 6a*), the hemorrhagic tissue damage was restricted to the middle and lower lobes of the right and the lower lobe of the left lung (*Figure 6b*). Vasculitis-like features were observed in patients 2, 3, 4, 8, and 9 with sporadic mild lymphoplasmatic cellular infiltrates around the pulmonary artery branches (*Figure 6c,d*). However, in patient 7, a strong lymphocyte-predominant intra-alveolar infiltrate was found (*Figure 6e,f*). In particular, megakaryocytes were sometimes detectable in alveolar capillaries (*Figure 6g,h*).

A common histological feature in all patients was the loss of the follicular architecture of the lymph nodes due to architectural changes (*Figure 7a*). In the bone marrow of patient 9, the highest viral loads were found, and significant hemophagocytosis was detectable by microscopy (*Figure 7b*). Interestingly, a correlation between a high viral load and tissue damage, as seen in the lung, was not found in cardiac or aortic tissues. The cardiac tissues sometimes showed pre-existing changes (fibrosis and chronic ischemic damage), but no severe damage, inflammation or necrosis of cardiomyocytes was found (*Figure 7c*). Sometimes an increase in cellularity in the otherwise unremarkable cardiac tissue was seen (e.g. in patient 1), which was suggestive of an activated cardio-mesenchyme

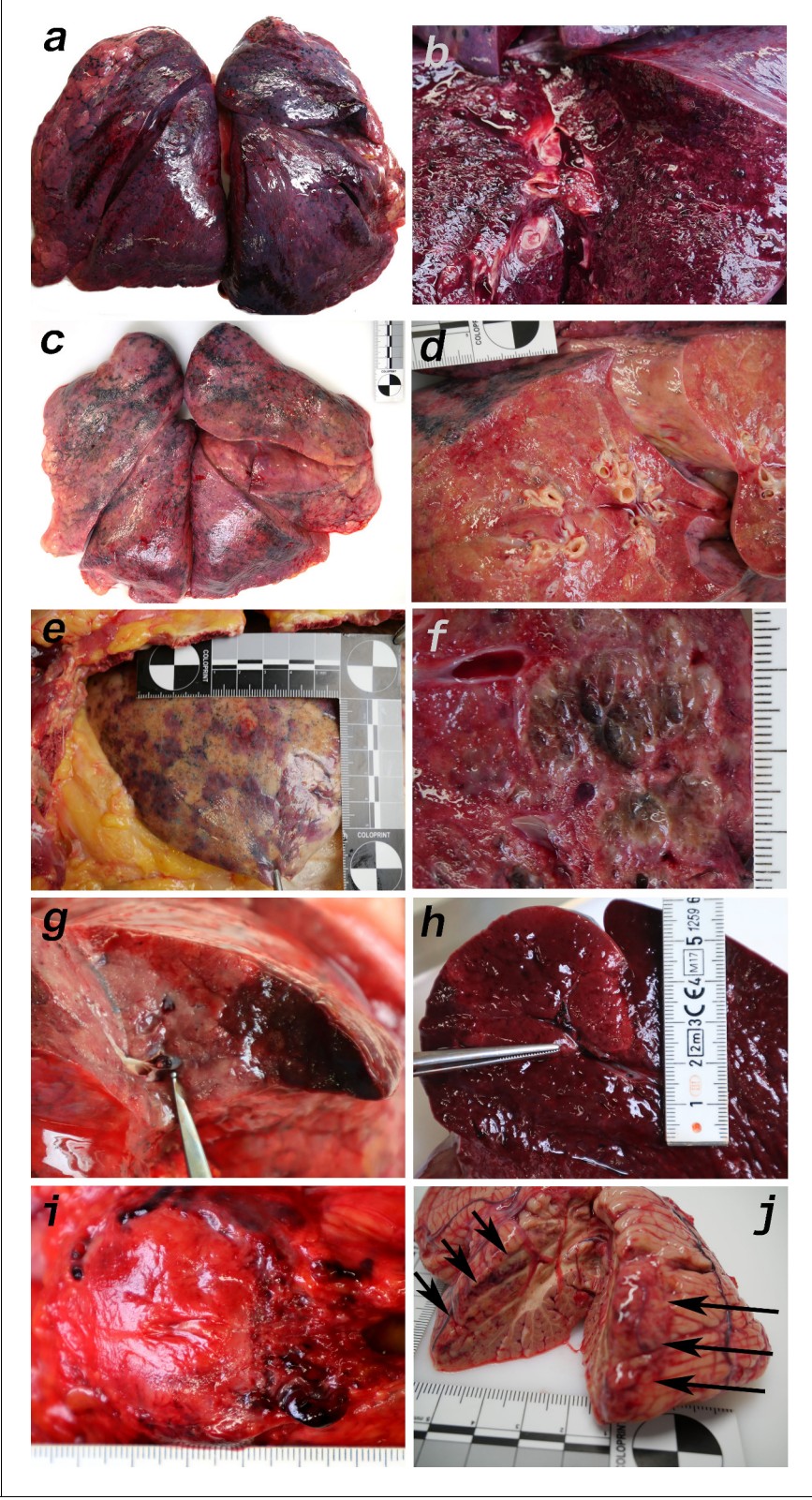

**Figure 4.** Macromorphology findings of COVID-19 patients. (**a**) Pneumonectomy of patient one showed strong congestion with liquids and hemorrhages. The tissue consistency was fragile. (**b**) Cut surface of lung tissue in higher magnification as shown in (**a**). The pleura shows further hemorrhages. (**c**) Pneumonectomy of patient seven showed a more solid lung tissue without congestion. The tissue consistency was very firm. (**d**) Cut surface of lung tissue in higher magnification as shown in (**c**). Lung tissue was retracted adjacent to the bronchus. (**e**) Pale pleura visceralis of the lung of

*Figure 4 continued on next page*

*Figure 4 continued*

patient 6 with disseminated hemorrhages and signs of disturbed ventilation. (**f**) Nodular transformation of lung tissue as phenomenon of fungal superinfection in patient 3. (**g**) Hemorrhagic lung infarct in patient 4 due to a thrombembolus in a pulmonary artery branch. (**h**) Anemic spleen infarct due to a clotted small artery in patient 4. (**i**) Fulminant stasis and thromboses in the periprostatic plexus in patient 4. (**j**) Cerebellar infarction (hemorrhagic) in patient 9.

(*Figure 7d*). The large vessels were unremarkable as well. *Figure 6e* demonstrates a section of the thoracic aorta of patient 5 obtained from the same anatomical location as the sample that tested highly positive for vRNA. Neither the intima (asterisk) nor the upper tunica media displayed any inflammatory cells or tissue damage. In the colonic mucosa, strong signs of epithelial damage were not visible by light microscopy (e.g. patient 3; *Figure 7g*), and the histology of the pancreas was well preserved (e.g. patient 9; *Figure 7h*).

## Discussion

In clinical practice, many critically ill COVID-19 patients show multiple organ involvement in addition to lung failure, in particular vascular dysfunction, including thrombosis and/or impaired microcirculation (*Ji et al., 2020*). A dysregulated immune response was observed, starting with a phase of immunosuppression followed by a proinflammatory phase and then a cytokine storm (*Li et al., 2020*). The cytokine storm may play an important role in COVID-19, which supports the hypothesis that COVID-19 could have a strong immunopathological component.

Some aspects of the viral pathogenesis and the toxicity of the novel virus SARS-CoV-2 are known based on previous studies of SARS-CoV (*Qian et al., 2013*). The virus can infect nasal mucous cells, pneumocytes and alveolar macrophages. ACE2 is the main receptor for the cellular binding process (*Bar-On et al., 2020*). Since the ACE2 receptor is expressed in cells in addition to those in the respiratory system (*Jia et al., 2005*), it is reasonable to assume that other organ systems can also be targeted by SARS-CoV-2.

Quite a few morphological studies have been published so far. Some are single case reports based on necropsies of the lung, liver and heart (*Xu et al., 2020*), partial autopsies of the thoracic cavity (*Karami, 2020*) or full autopsies (*Aguiar et al., 2020*). Some involve small (n = 2–3) case series based on surgical lung resectates (*Tian et al., 2020a*) or on full autopsies (*Ding et al., 2003*; *Sekulic et al., 2020*; *Barton et al., 2020*; *Varga et al., 2020*). Larger case studies (n = 4-28) have focused on single organs such as the lungs (*Ackermann et al., 2020*), spleen (*Xu, 2020*), or kidney (*Puelles et al., 2020*) or on the heart and lungs (*Buja et al., 2020*; *Fox et al., 2020*) or liver, heart and lungs (*Tian et al., 2020b*); others have reported comprehensive organ findings obtained by minimally invasive sampling (*Duarte-Neto et al., 2020*) or full autopsies (*Menter et al., 2020*; *Schaller et al., 2020*; *Wichmann et al., 2020*; *Edler et al., 2020*; *Bösmüller et al., 2020*; *Bradley et al., 2020*; *Remmelink et al., 2020*; *Skok et al., 2020*; *Hanley et al., 2020*). Several of the aforementioned autopsy studies reported viral loads in selected organs and tissues (*Sekulic et al., 2020*; *Puelles et al., 2020*; *Tian et al., 2020b*; *Menter et al., 2020*; *Wichmann et al., 2020*; *Bösmüller et al., 2020*; *Bradley et al., 2020*; *Remmelink et al., 2020*; *Hanley et al., 2020*). The postmortem interval between death and autopsy was either not reported (*Tian et al., 2020b*; *Bradley et al., 2020*; *Skok et al., 2020*) or was > 48 hr (*Lax et al., 2020*), 11–84.5 hr (*Menter et al., 2020*), 72–96 hr (*Remmelink et al., 2020*), 1–5 days (*Wichmann et al., 2020*), 4 days (on average with a maximum of 12 days) (*Edler et al., 2020*), or 6 days (*Hanley et al., 2020*). The small number of patients in these studies is consistent with the sample size of many other studies but is nevertheless a limitation. However, we analyzed a considerably larger number of organs and tissues than the other groups. To our knowledge, the present study is the only one to date that has measured viral loads in a wide variety of organs and tissues by obtaining and processing 61 samples per patient. The present study is the only study so far that has focused on keeping the postmortem interval as short as possible to avoid bias due to the degradation of SARS-CoV-2 virus particles, SARS-CoV-2 RNA and tissue ultrastructure. Regarding vRNA degradation, the values reported by *Puelles et al., 2020* show comparatively low viral loads among their cases, even in the lungs, which is the primary target of SARS-CoV-2. *Wichmann et al., 2020* reported the highest values in the lungs of $1.2 \times 10^4$ to $9 \times 10^9$ copies/ml, while the highest values in our study were $\sim 10^7$

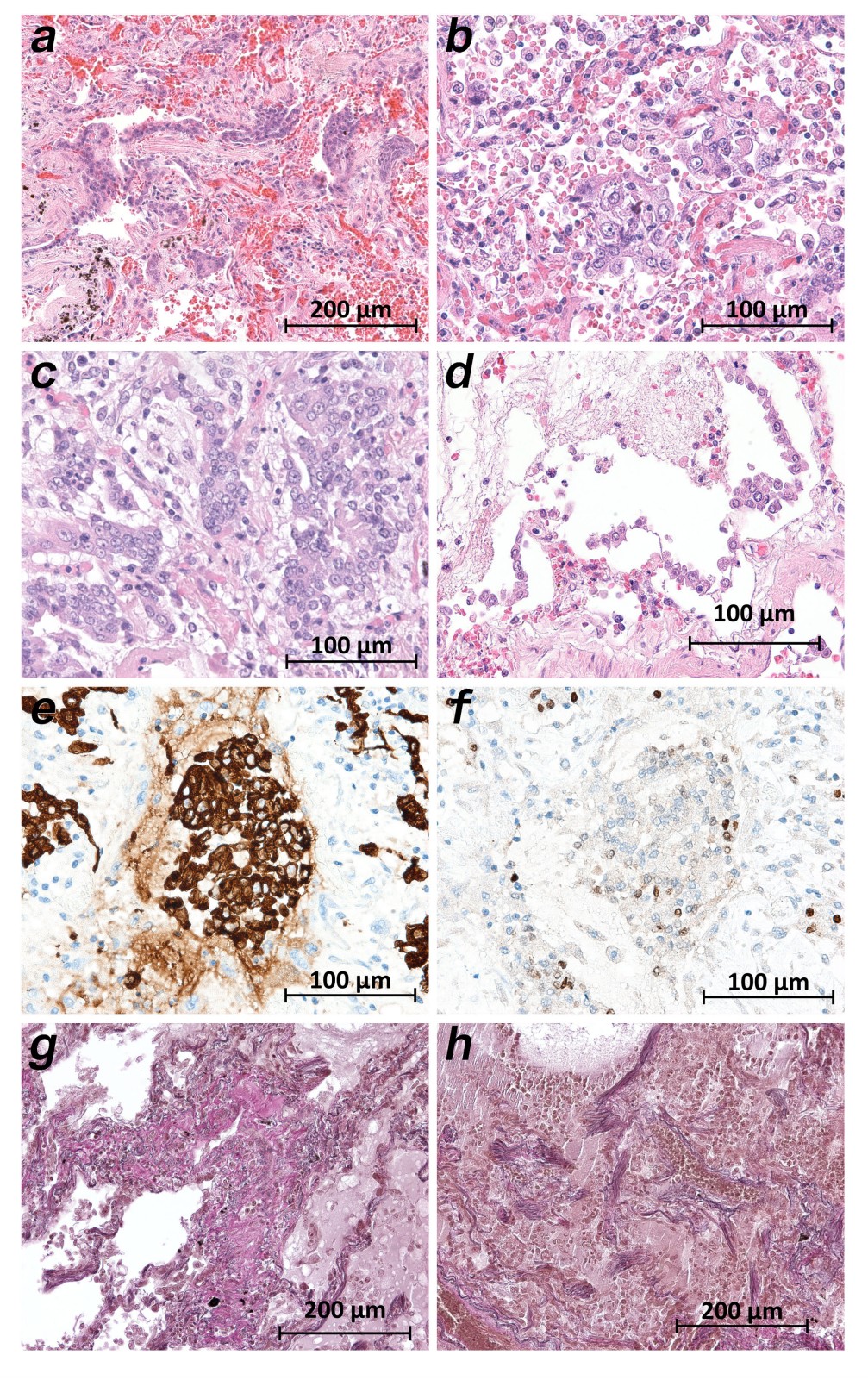

**Figure 5.** Micromorphology lung findings of COVID-19 patients. (a) Destroyed lung tissue with intraalveolar hemorrhagia and aggregates of prominent epithelial cells resembling squamous metaplasia (patient 1; HE). (b) Strong architectural damage of lung alveolar tissue with disruption of the epithelial barrier and intraalveolar accumulation of enlarged cells with prominent nuclei and visible nucleoli. Initial syncytial pattern is given (patient 2; HE). (c) Lung tissues with multinucleated giant cells admixed with only few lymphocytes (patient 4; HE). (d) Alveolar unit with band-like desquamation of the

*Figure 5 continued on next page*

*Figure 5 continued*

alveolar epithelial cells in the alveolar space partially filled with liquids, erythrocytes and few lymphocytes (patient 5; HE). (**e**) Multinucleated giant cell in an alveolar space is strongly positive for keratins (patient 4; immunostaining AE1/3). (**f**) Serial section of (**e**), the multinucleated giant cell after immunostaining against TTF1 (patient 4; immunostaining TTF1). (**g**) Lung tissue with interstitial fibrosis (patient 8; EvG). (**h**) Lung tissue with interstitial and intraalveolar fibrosis (patient 8; EvG).

vRNA copies/ml. From *Table 2* it becomes clear that the duration of post-mortem intervals in our study did not substantially influence the vRNA copy numbers. The importance of obtaining multiple samples from within one organ is emphasized by the results of patient 10 (*Figure 1—figure supplement 1*), in which only two of seven samples from the heart were positive. If only one sample had been obtained, the result of viral detection could have been a false negative. Based on the mapping of SARS-CoV-2 RNA throughout the whole human body, we were able to correlate the viral loads in many organs and tissues with the macro- and micromorphology. TEM investigations of the lung samples revealed the presence of morphologically intact virus particles in the tissue, which was in line with the vRNA mapping, which showed the highest viral loads in the lungs. The morphologically intact virus particles were located in lung fibrocytes. In agreement with data published by *Varga et al., 2020*, viral inclusion bodies were detectable using TEM. In lung tissues, the morphology of the viral particles was clearly observed (*Figure 2*), whereas in other tissues such as liver, heart, and intestine, viral inclusions were not visible by TEM. The loss of structural hallmarks could be due to postmortem cell and tissue turnover and the reduced integrity of virus particles during TEM-related preparations. It has to be stressed that the molecular detection of the virus does not depend on the presence of morphologically intact virus particles.

We detected the highest viral loads and the most severe tissue damage in the lungs. The lung samples of all patients showed large cells, sometimes multinucleated giant cells, that were similar to the giant cells described in cases of respiratory syncytial virus (RSV) infection. The preliminary immunostaining pattern of the enlarged cells indicated that they represented affected pneumocytes. Squamous (metaplastic) large cells and clusters of giant cells have been reported by most morphological studies, with one exception (*Ackermann et al., 2020*). The remaining findings from the lung samples agree very well with the findings of other groups, especially the data of *Duarte-Neto et al., 2020*. The strict topological correlation of viral loads and histopathological damage is emphasized by the results in patient 2. The samples from the upper lung lobes showed normal, unremarkable tissue (*Figure 6a*) corresponding to a negative viral test result, while the samples from the lower lung lobes revealed severe tissue damage corresponding to high and moderate viral loads (*Figure 6b*).

All patients who died due to COVID-19 (patients 1–10) had viral RNA in at least some samples of the lymphatic tissue. Lymphatic tissue with a topological relationship with the respiratory tract (e.g. tonsils, cervical lymph nodes, and hilar lymph nodes) was more likely to be positive overall than lymphatic tissue without such a topological relationship (e.g. mesenteric lymph nodes, spleen, and appendix). One remarkable finding in the lymph node samples of all patients was the loss of the follicular structure (*Figure 7a*). Atrophy of lymphatic tissue has been described in association with SARS-CoV infection by *Gu et al., 2005* and discussed as a crucial determinant of disease outcome by *Perlman and Dandekar, 2005*. Lymphocyte depletion has also been reported by *Hanley et al., 2020*. The spleen was positive in patients 1, 3, 4, 5, and 9, who presented with the micromorphology of early lung damage, and negative in patients 6, 7, 8, and 10, who presented with the micromorphology of later lung damage or who did not die of COVID-19 pneumonia (patient 2). Further interpretation of the viral loads in the appendix is futile since the appendix showed age-related and chronic pathologic changes accompanied by the loss of lymphatic tissue.

Viral loads in the cardiac tissue were moderate to very low and systematically (in all samples) detected in the patients with early lung damage, while patients with later lung damage displayed high viral loads only sporadically or not at all. The cardiac histology of the left ventricle, anterior wall, and basal portion in patient one with a moderate viral load is presented in *Figure 7c,d*. Except for the activation of mesenchymal cells, which requires further investigation, the histology was unremarkable. In accordance with *Buja et al., 2020*, in patient 1, we also observed pericarditis and multifocal acute injury of cardiomyocytes, for example, myocardial contraction band necrosis, which is frequently observed in critically ill patients under catecholamine therapy.

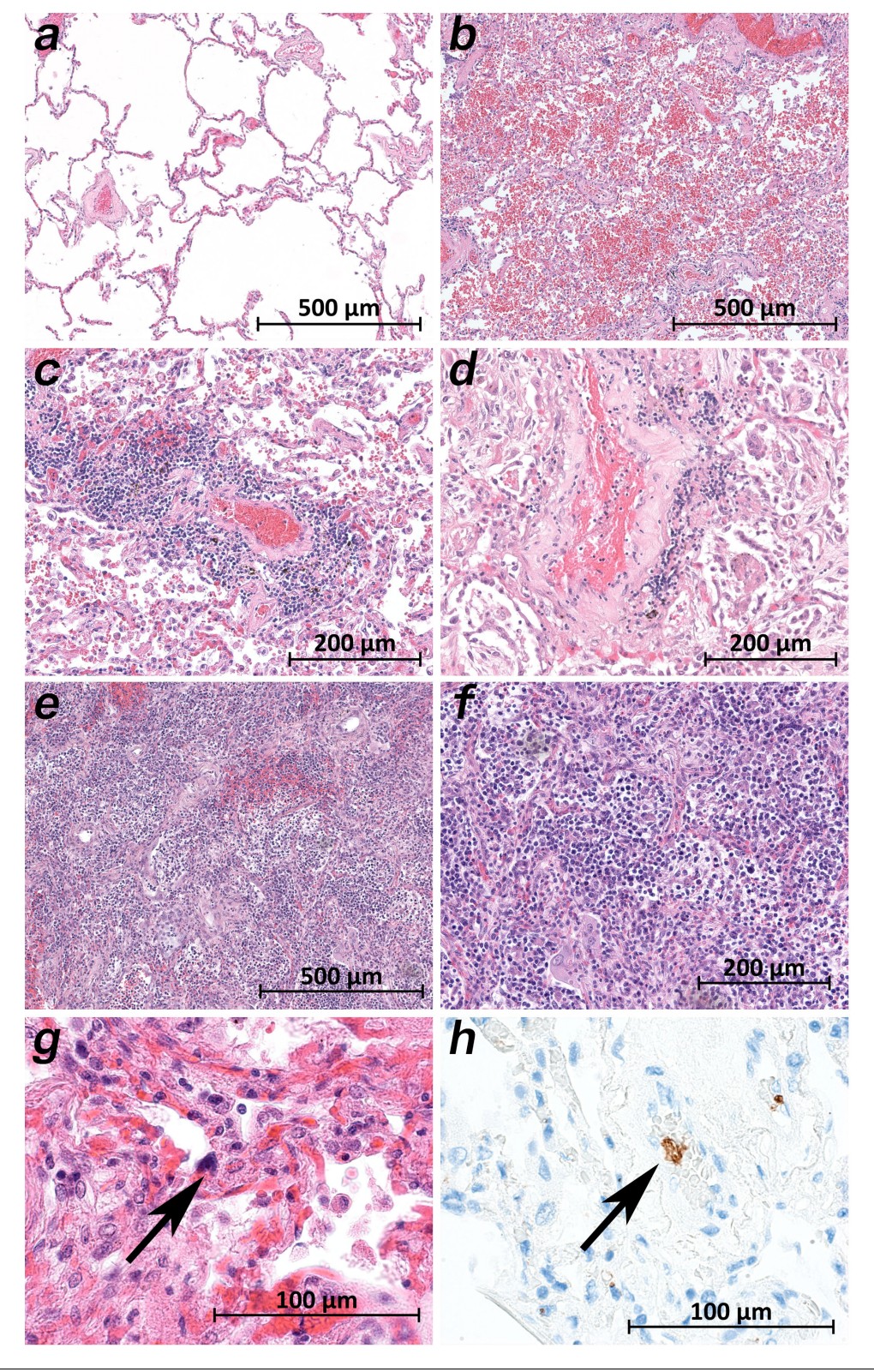

**Figure 6.** Micromorphology lung findings of COVID-19 patients. (**a**) Lung tissue with minimal emphysematous changes derived from the upper lobes without detectable viral loads (patient 2; HE). (**b**) Severe hemorrhagic pneumonia in specimens from the lower lobes with high viral loads (patient 2; HE). (**c**) Vasculitis-like changes around pulmonary artery branches (patient 4; HE). (**d**) Damaged lung tissue with hemostasis and inflammatory changes adjacent to the pulmonary artery branch (patient 4; HE). (**e**) Strong lymphocytic-predominant infiltration of lung tissues with hemorrhagic and interstitial

*Figure 6 continued on next page*

*Figure 6 continued*

edema (patient 7; HE). (**f**) Higher magnification of the lymphocytic-predominant infiltrate (patient 7; HE). (**g**) Lung tissue with a large nucleated cell in an alveolar capillary, suggestive for a megakaryocyte (arrow; patient 9; HE). (**h**) The same tissue after immunostaining against CD61 (patient 9; immunostaining CD61).

The viral loads in the samples from the vascular tissue (aorta and pulmonary artery) followed a similar distribution pattern depending on the stage of lung damage but were higher compared to those in the cardiac tissue. The unremarkable histology of the thoracic aorta of patient 5 with a high viral load is presented in *Figure 7e,f*. The conclusion by *Varga et al., 2020* that SARS-CoV-2 induces endothelitis cannot be comprehended.

Interestingly, the presence of detectable vRNA in the gastrointestinal tract was variable. The very high viral loads in patient nine throughout the upper gastrointestinal tract as well as in the small and large bowels were noticeable. The histology of the corresponding tissue samples was unremarkable. According to the clinical documentation, patient 9 did not exhibit any gastrointestinal symptoms.

Viral RNA could also be detected in low to very high amounts in the samples from the endocrine organs, urinary tract, nervous system, and reproductive system. Interestingly, the samples of patients 1, 2, 6, and 7, who were treated with lopinavir/ritonavir, tested negative. Our results support the findings of *Remmelink et al., 2020*. of nonspecific postmortem organ findings despite multiorgan viral spread.

The same distribution pattern among the patients was observed regarding viral RNA in blood. Apart from patients 8–11, who were not subjected to intensive care (patients 8–10) or did not die of COVID-19 (patient 11), it is noticeable that among the patients who received intensive care prior to death (patients 1–7), only patients 3, 4, and 5 tested positive for vRNA in blood, while patients 1, 2, 6, and 7 tested negative. The latter patients were treated with lopinavir/ritonavir, so the effect of antiviral medication on preventing viremia may be indicated. In our study, 4 of 11 patients were treated with an antiviral medication. None of these four patients showed the presence of vRNA in the blood, but one patient showed the presence of vRNA in the bone marrow (patient 1). The application of antiviral drugs is currently being investigated in many studies. However, it has not yet been shown that a significant effect can be achieved by their application (*Cao et al., 2020*). However, drugs are often used in severe cases in the ICU based on controversial recommendations (*Meini et al., 2020*). Our data could indicate that lopinavir/ritonavir leads to a reduction in viremia. However, our group size is too small for such a statement, as we could not detect vRNA in the blood, not even in untreated patients in some cases.

The patients with vRNA in the blood also showed vRNA in the bone marrow. Patients 1, 8, and 9 were negative for vRNA in the blood but positive in the bone marrow. Patient 9 showed the highest viral loads by far in the bone marrow. The histology of the bone marrow, apart from hypercellularity, a left shift and an increased number of megakaryocytes, showed a significant amount of hemophagocytosis (*Figure 6b*). Hemophagocytosis was also reported by *Hanley et al., 2020* and is a morphological feature of macrophage activation syndrome (MAS) or hemophagocytic lymphohistiocytosis (HLH) (*Crayne et al., 2019*; *Al-Samkari and Berliner, 2018*). The clinical characteristics of COVID-19, including very high ferritin levels and very high levels of proinflammatory interleukins, resemble those of MAS and HLH (*Colafrancesco et al., 2020*) and have already led to several therapeutic attempts (*Dimopoulos et al., 2020*). Further studies are needed to clarify this aspect.

A positive test result based on qRT-PCR can be determined with the Ct-value. Due to the further spread of the pandemic, clinicians have discussed the release of this value as part of the test result, as the viral load has been identified as an important prognostic indicator (*Magleby et al., 2020*). The authors suggest using the Ct-value to identify patients with a high risk for severe clinical courses. Further studies also indicate that the viral load may serve as a surrogate marker of infectivity associated with mortality rates (*Faico-Filho et al., 2020*).

However, the College of American Pathologists reminds us to be cautious in interpreting the results (*Rhoads et al., 2020*; *Jaafar et al., 2020*). As a multitude of intra- and interlaboratory factors influence detection, the Ct-value has to be critically evaluated. Due to the relatives' necessary consent before the autopsy, the specimen collection is only possible with a time delay in clinical routine. Our study did this particularly urgently, but there are still differences in the time of sample

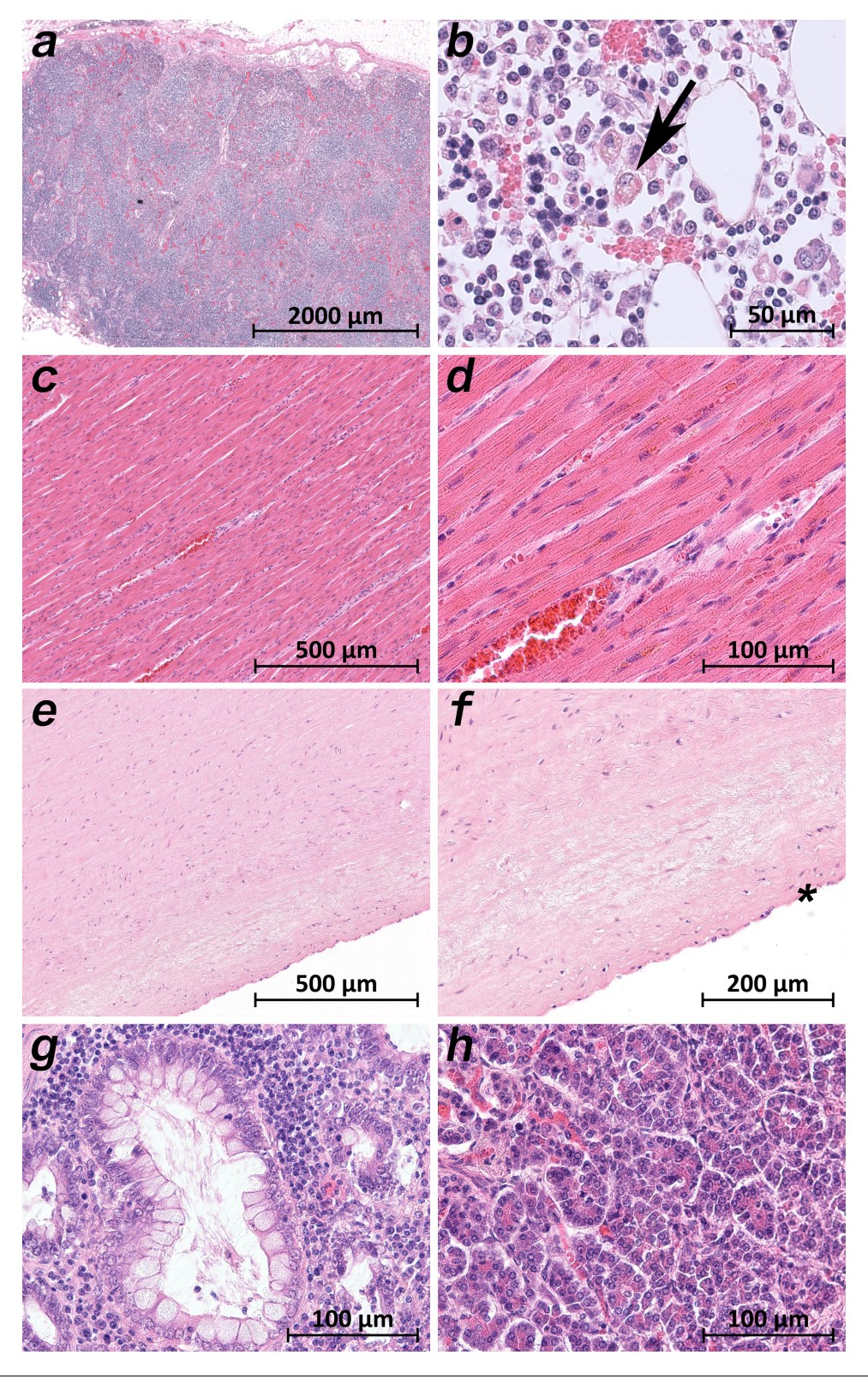

**Figure 7.** Extrapulmonary micromorphology findings of COVID-19 patients. (a) Overview of a mediastinal lymphnode with some nodular aggregates of lymphocytes, but destroyed lymphofollicular structures (patient 5; HE). (b) Bone marrow with prominent hemophagocytosis (arrow) and maturating cells of the hematopoiesis (patient 9; HE). (c) Myocardial tissue of the left ventricle (patient 1; HE). (d) Myocardial tissue of the left ventricle in higher magnification with a minimal increase in cellularity indicating for an activated cardiomesenchyme (patient 1; HE). (e) Thoracic aorta with a low number of

*Figure 7 continued on next page*

Figure 7 continued

non-inflammatory nucleated cells in an unsuspicious matrix (patient 5; HE). (f) Tissue from the thoracic aorta in higher magnification. The endothelium is labeled with an asterisk (patient 5; HE). (g) Colon mucosa with a crypt lined by goblet cells and enterocytes without any strong intraepithelial inflammation (patient 3; HE). (h) Exocrine pancreas tissue with structural intact acini without inflammatory cells (patient 9; HE).

collection. This is an important limitation of our study and must be noted when considering the results. However, the subsequent processing of the samples was performed identically. We use the same methods for the RNA-extraction as well as the same qRT-PCR method.

Another important reason for the detection of Ct-values is the statement about infectiousness. Jaafar et al. showed that the culture of the virus is successful up to a Ct-value of 25 (70%). At a Ct-value of 30, this value drops to 20%, and values above 35 as associated with only a low likelihood (3%) of the possibility of culture (*Jaafar et al., 2020*). These values show a correlation to a certain extent, although it should be noted that the culture of viruses from patient material is generally challenging and that no absolute indication can be made based on a Ct-value for this purpose exclusively.

To further elucidate the immunological host response, we measured IL-6 and IL-8 in postmortem serum samples. The serum levels of interleukin six were significantly elevated in all patients, including patient 11, who died of multiple organ failure following ileus. The serum levels of IL-8 were also significantly elevated. In accordance with other autopsy studies (*Wichmann et al., 2020*; *Lax et al., 2020*), we observed thromboembolic events in 4 of the 11 patients. Patient 2 died of pulmonary embolism, and patient 4 suffered multiple lung and spleen infarctions due to venous thromboses. Patients 5 and 10 presented with sporadic emboli in the lung histology. In conjunction with the general impairment of the microcirculation, which was histologically visible as homogenous eosinophilic sludge in the small arterioles, capillaries and venules in multiple organ samples from patients 1, 8 and 9, 7 (of 11), several patients suffered hypercoagulation. Some studies hypothesize that SARS-CoV-2 can induce noncoordinated reactions between the coagulation and fibrinolysis systems that result in hypercoagulation and hemorrhage (*Ji et al., 2020*). The levels of the measured pro-thrombotic factors were almost all significantly elevated in all patients.

In summary, we presented an autopsy series of 11 patients with COVID-19. The autopsies were performed in the early postmortem interval to avoid bias due to the degradation of vRNA, virus particles, and tissue structures. SARS-CoV-2 RNA could be detected in very high to high amounts in the lungs and in very high to very low amounts in the lymphatic tissue. TEM visualized SARS-CoV-2 particles in the lung tissue. Viral loads and histological tissue damage were strongly correlated in the lungs even at the organ level (patient 2). Histological structure changes were also present in the lymph nodes (atrophy and loss of follicles). High viral loads were detected in many other tissue samples from different extrapulmonary organs and tissues without evident tissue damage based on light microscopy.

## Materials and methods

### Autopsies and postmortem sampling

The study was approved by the local ethical board (registration no.: 2020–1773). Complete autopsies (inspection of cranial, thoracic and abdominal cavities plus dissection of all internal organs and their surrounding anatomical structures) of 11 patients with SARS-CoV-2 infection (proven by naso-pharyngeal swab testing during hospitalization) and a clinical diagnosis of COVID-19 were included in this study. As soon as possible after death, the closest relatives were contacted, who gave their informed consent. The autopsies in this study were performed 1.5–15 hr (mean 5.6 hr) postmortem, and the organs were dissected directly without prior fixation. The same team including two experienced forensic pathologists conducted all autopsies. The lungs, with expected high viral loads, were removed first and dissected last to avoid the transfer of vRNA to other organs/tissues. At each autopsy, a total of up to 61 native and nonfixed samples (five locations in the nervous system, 14 in the respiratory tract with double sampling of the lungs, 10 in the cardiovascular system, 12 in the gastrointestinal tract, 3 in the urinary tract, 4 in the reproductive system, 2 in the endocrine system, 6 in the lymphatic system, 2 in hematological tissues, and 3 in abdominal skin, abdominal

subcutaneous tissue and the musculus psoas major) were collected after rinsing them in clean tap water. Tissue samples were transferred for virological processing immediately after autopsy; blood samples were centrifuged to obtain serum. Samples from the same anatomical locations were fixed in 5% buffered formalin solution for comparative histopathological analysis, and selected samples were placed in a special fixative (see below) for TEM. Since the correct electron microscopic identification of virus particles in tissue obtained by autopsy can be challenging (*Goldsmith et al., 2020*), all possible precautions were taken. That included maintaining the shortest possible post-mortem time, the very meticulous handling of the small pieces of tissues, the exclusive use of TEM-grade, fresh chemicals, the strict adherence to the schedule and the execution of tasks by trained and experienced staff. In addition, a reference sample was prepared from a cell culture (Vero-76 cells) infected with SARS-CoV-2 to be able to clearly distinguish the virus findings from normal cytoplasmic structures such as parts of the endoplasmic reticulum, Golgi apparatus, coated vesicles, or artefacts caused by degradation.

## SARS-CoV-2 RNA detection

All tissues were homogenized in RPMI medium by using the FastPrep-24 5G Instrument (MP Biomedicals, Schwerte, Germany), and disposable homogenizer beads (Zymo Research Bashing Bead Lysis Tubes, Freiburg, Germany) were used to avoid contamination. We placed 200 mg of each tissue/organ in 1000 µl RPMI-1640 (Roswell Park Memorial Institute, Thermo Fisher Scientific GmbH, Dreieich, Germany). After a centrifugation step (2 min, 12,000 rpm), the supernatants were collected for the determination of the viral load. RNA extraction was performed by using the QIAcube RNeasy Viral Mini Kit (Qiagen, Hilden, Germany) according to the manufacturer's guide. qRT-PCR was performed using RIDAgene (r-biopharm, Darmstadt, Germany) with the Rotor-Gene Q (Qiagen, Hilden, Germany) to detect the E-gene of SARS-CoV-2 by determining the cycle threshold (Ct) value. The RNA standard curve, prepared by amplification of the positive control with the RIDAgene (r-biopharm, Darmstadt, Germany) kit, was applied for quantification. SARS-CoV-2 RNA is represented as the decadic logarithm of the number of copies/ml. The following scale was applied: very high (>$10^4$ copies/ml), high ($10^3$–$10^4$ copies/ml), moderate ($10^2$–$10^3$ copies/ml), low ($10^1$–$10^2$ copies/ml), and below the detection limit (bdl).

## Detection of inflammatory and thrombotic parameters

For the measurement of proinflammatory cytokines and coagulation parameters, a Legendplex Human Thrombosis Panel (13-plex) (BioLegend, San Diego, CA, USA) was used. Twenty-five microlitres of each serum sample was transferred in duplicate into a 96-well filter plate, and the Legendplex panel was performed by following the manufacturer's instructions. The samples were measured on the same day on a flow cytometer (BD, Accuri), and the protein amount was calculated by comparison to a standard curve. Serum samples from five healthy volunteers without any signs of infection were age-correlated and analysed as a control.

## Histopathological analysis

After fixation for at least 24 hr in 10% neutral buffered formalin, the tissue samples were dehydrated in a graded series of ethanol and xylene, mounted in paraffin and cut into 3-µm-thick sections. In addition to hematoxylin and eosin (HE), Elastica-van-Gieson (EvG), Berlin Blue (Fe), periodic acid Schiff stain (PAS), Alcian Blue-periodic acid Schiff stain (abPAS), Giemsa, Gomori Trichrome, and Kongo Red stain were used by following routine protocols. For immunohistochemistry, the following antibodies were used: AE1/3 (Dako/IR053), TTF-1 (Dako/IR056), CK7 (Dako/IR619), CK5/6 (Dako/IR780), p40 (Zytomed/MSK097), Ki67 (Dako/IR626), CD68 (Dako/M0876), CD61 (Dako/M0753), CD31 (Dako/IR610), CD34 (Dako/IR623), ASMA (Dako/IR611), CD3 (Dako/IR503), CD20 (Dako/IR604), MUM1 (Dako/IR644), collagen IV (Dako/M0785), and tenascin (Chemicon/MAB19101). All immunostaining were performed with the Dako Omnis immunostainer (Agilent) by following routine procedures. The sections were examined microscopically (Axio Imager. M2, Carl Zeiss Microscopy GmbH), and representative photographs were obtained (Axiocam 506 color, Carl Zeiss Microscopy GmbH; ZEN 2.6 (blue edition), Carl Zeiss Microscopy GmbH).

## Transmission electron microscopy

During each autopsy, several small pieces of lung tissue (2 mm cubes) were immediately fixed with freshly prepared modified Karnovsky fixative (4% w/v paraformaldehyde and 2.5% v/v glutaraldehyde in 0.1 M sodium cacodylate buffer, pH 7.4) for 24 hr at room temperature. After washing three times for 15 min each with 0.1 M sodium cacodylate buffer (pH 7.4), the tissue was further cut into 1 mm cubes and postfixed with 2% w/v osmium tetroxide for 1 hr at room temperature. During the subsequent dehydration in an ascending ethanol series, poststaining with 1% w/v uranyl acetate was performed. Afterwards, the samples were embedded in epoxy resin (Araldite) and sectioned using a Leica Ultracut S (Leica, Wetzlar, Germany). Based on the examination of semi-thin sections, regions of interest of approximately 500 µm x 500 µm in size were selected and trimmed. Finally, the ultrathin sections were mounted on filmed Cu grids, post-stained with lead citrate, and studied in a transmission electron microscope (EM 900, Zeiss, Oberkochen, Germany) at 80 kV and 3000–50,000x magnification. For image recording, a 2K slow scan CCD camera (TRS, Moorenweis, Germany) was used.

## Acknowledgements

We thank Jenny Pfeifer, Nico Möller, Cornelia Jacob and Christine Weiler with her fellow technicians for their excellent technical support. The authors also thank Anika Hopf (Center for Electron Microscopy) for her excellent work in sample preparation.

## Additional information

### Funding

| Funder | Grant reference number | Author |
| --- | --- | --- |
| Interdisziplinäres Zentrum für Klinische Forschung, Universitätsklinikum Jena | ACSP02 | Stefanie Deinhardt-Emmer |
| Carl Zeiss Foundation | | Christina Ehrhardt |
| German Research Foundation | NUM-COVID 19 | Stefanie Deinhardt-Emmer |
| Fund of the Thueringer Universitaets- und Landesbibliothek Jena | 433052568 | Christina Ehrhardt |
| German Research Foundation | Organo-Strat 01KX2021 | Stefanie Deinhardt-Emmer |

The funders had no role in study design, data collection and interpretation, or the decision to submit the work for publication.

### Author contributions

Stefanie Deinhardt-Emmer, Conceptualization, Data curation, Software, Formal analysis, Supervision, Funding acquisition, Validation, Investigation, Visualization, Methodology, Writing - original draft, Project administration, Writing - review and editing; Daniel Wittschieber, Conceptualization, Data curation, Software, Formal analysis, Supervision, Validation, Investigation, Visualization, Methodology, Writing - original draft, Project administration, Writing - review and editing; Juliane Sanft, Karoline Frieda Haupt, Vanessa Vau, Clio Häring, Data curation, Formal analysis, Investigation, Methodology; Sandra Kleemann, Stefan Elschner, Andreas Henke, Investigation, Methodology; Jürgen Rödel, Resources, Writing - review and editing; Christina Ehrhardt, Funding acquisition, Writing - review and editing; Michael Bauer, Resources, Data curation; Mike Philipp, Data curation, Investigation; Nikolaus Gaßler, Resources, Supervision, Investigation, Methodology, Writing - review and editing; Sandor Nietzsche, Investigation, Methodology, Writing - review and editing; Bettina Löffler, Conceptualization, Resources, Data curation, Software, Supervision, Validation, Visualization, Writing - original draft, Project administration, Writing - review and editing; Gita Mall, Conceptualization, Resources, Data curation, Software, Formal analysis, Supervision, Validation, Investigation, Visualization, Methodology, Writing - original draft, Project administration, Writing - review and editing

## Author ORCIDs
Stefanie Deinhardt-Emmer (iD) https://orcid.org/0000-0003-4495-4052
Michael Bauer (iD) https://orcid.org/0000-0002-1521-3514

## Ethics
Human subjects: The study was approved by the local ethical board (registration no.: 2020-1773). Complete autopsies (inspection of cranial, thoracic and abdominal cavities plus dissection of all internal organs and their surrounding anatomical structures) of 11 patients with a SARS-CoV-2 infection (proven by naso-pharyngeal swab testing during hospitalization) and the clinical diagnosis of COVID-19 were included in this study. As soon as possible after death the closest relatives were contacted and gave their informed consent.

## Decision letter and Author response
Decision letter https://doi.org/10.7554/eLife.60361.sa1
Author response https://doi.org/10.7554/eLife.60361.sa2

## Additional files

### Supplementary files
• Transparent reporting form

### Data availability
RNA data generated or analysed during this study are included in the supporting files.

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
