## [Decision Letter]

**Acceptance summary:**

This study of 11 COVID-19 full autopsy cases, describes histopathology in relation to TEM detection of intact virus particles in lung, PCR virus RNA detection in other organs, and cytokines and coagulation markers in serum. The study represents a thorough body of work that may enrich our insight into COVID-19 pathogenesis. The main message is that COVID-19 is a systemic disease as determined by presence of virus RNA, yet not related to tissue damage outside the lungs. This adds to the field in its extensiveness of testing within each patient. A side message is that COVID-19 in selected cases may show a loss of follicular structure.

**Decision letter after peer review:**

Thank you for submitting your article "Early postmortem mapping of SARS-CoV-2 RNA in patients with COVID-19 and correlation to tissue damage" for consideration by *eLife*. Your article has been reviewed by two peer reviewers, and the evaluation has been overseen by a Reviewing Editor and a Senior Editor. The following individual involved in review of your submission has agreed to reveal their identity: Katrien Grunberg (Reviewer #2).

The reviewers have discussed the reviews with one another and the Reviewing Editor has summarized the essential points to help you prepare a revised submission.

Summary:

This study of 11 COVID-19 full autopsy cases, describes histopathology in relation to TEM detection of intact virus particles in lung, PCR virus RNA detection in other organs, and cytokines and coagulation markers in serum. The sample size is small but relevant and sufficient for a descriptive study. The study represents a thorough body of work that may enrich our insight into COVID-19 pathogenesis. The main message is that COVID-19 is a systemic disease as determined by presence of virus RNA, yet not related to tissue damage outside the lungs. This adds to the field in its extensiveness of testing within each patient. A side message is that COVID-19 in selected cases may show a loss of follicular structure.

Revisions:

1) All the data reported in the first paragraph of the Introduction must be updated.

2) Please remove "extraordinary" from the phrase "measured SARS-CoV-2 RNA in an extraordinary high number of samples".

3) Given the statements made by the authors "Contrary to recommendations of the pathological professional societies not to perform autopsies earlier than 24 hours post-mortem and to store the organs in formalin before dissecting them, the autopsies in this study were performed 1.5 – 15 h (mean 5.6 hours) post-mortem and the organs were dissected directly without prior fixation. The same voluntary team including two experienced forensic pathologists conducted all autopsies considering all necessary precautions and using the comprehensive personal protective equipment as recommended by professional societies"; it is necessary to mention the numerous national and European sources that dictate these precautionary rules.

4) The stains EvG, Fe, PAS, abPAS, must be preceded by the whole word (Elastica-van-Gieson, Berlinerblau, Periodic acid-Schiff stain, Alcianblau-PAS) before the acronym.

5) "Macroscopic signs of severe and extensive lung damage were found in all patients. In patients 1, 3-5, 9 and 10, the lungs with prominent hyperemia and edema displayed a fragility of the tissue". What does it mean?

6) I do not think it is appropriate to keep these descriptive self-satisfaction phrases in the text as the number of cases is still small. "The sample size of our case series can be compared to the larger autopsy case series from literature".

7) "multifocal acute injury of cardiomyocytes which is frequently observed in critically ill patients under catecholamine therapy". The authors must specify the type of cardiac damage by representing whether it is a coagulation myocytolysis or other.

8) Discussion paragraphs thirteen and fourteen can be easily removed.

9) Figure 4 is very trivial.

10) Figure 7C-D I do not understand the usefulness of showing a normal myocardium in c and I do not see any alteration of the myocardium in d.

11) The references must also be expanded for the autopsy case reports that have been published and that are not mentioned in this paper.

The main message is that COVID-19 is a systemic disease as demonstrated by presence of virus in organs outside the lungs. The description of tissue damage is mostly about lung. Apparently, a correlation viral load – tissue damage could not be established outside lung, while some lesions where present in other organs. The claim that these data allows the study of a correlation between viral load and tissue damage is not fully justified. A few limitations:

12) Quantification of viral load insufficiently explained. Materials and methods/Sars-cov2 detection quantification expressed as copies/ml: how did the investigators standardize for amount of tissue in medium

13) No TEM in other organs beside lungs. PCR detects all RNA, not necessarily intact viral particles, so unclear if this is just degraded virus, being transported from lung though the body.

14) The digestive tract and pancreas not exactly great for post mortem studies, even when the interval between exitus and autopsy is short. Subtle changes may be missed.

15) It is unclear from what area of organs exactly and whether lesional areas were specifically sampled or not. For example, was kidney cortex was properly sampled? So insight in viral tropism and where damage is to be expected is limited.

16) Role of antiviral drugs (potentially interfering with this correlation) is not discussed.

None of these limitations is discussed. Both parts of the conclusion that COVID-19 is 1 a systemic disease and 2 that there is no correlation with tissue damage outside the lung are in my view overstretched.

17) The authors present no (plan to study) data on interconnectedness of inflammation, thrombosis and infection/viral load in their Materials and methods. Correlation of IL-6, coagulation markers, presence of virus is unclear from results and figures. E.g. hemophagocytosis in bone marrow observed in one case and is discussed as possibly related to MAS and cytokine storm. Yet, no mention of IL-6. Hence, the presentations of any correlation comes across as a haphazard fishing expedition.

18) The conclusion ("Virus may be able to infect different cell types in different organs and tissues but may not be able to replicate in non-respiratory organs/tissues as efficiently as in the respiratory ones") is not based on the findings themselves hence not a conclusion but it may be discussed as a speculative explanation of the findings.

---

## [Author Response]

Revisions:1) All the data reported in the first paragraph of the Introduction must be updated.

Modification has been made. The data in the first paragraph have been updated.

2) Please remove "extraordinary" from the phrase "measured SARS-CoV-2 RNA in an extraordinary high number of samples".

Modification has been made. “Extraordinary” was removed from the sentence.

3) Given the statements made by the authors "Contrary to recommendations of the pathological professional societies not to perform autopsies earlier than 24 hours post-mortem and to store the organs in formalin before dissecting them, the autopsies in this study were performed 1.5 – 15 h (mean 5.6 hours) post-mortem and the organs were dissected directly without prior fixation. The same voluntary team including two experienced forensic pathologists conducted all autopsies considering all necessary precautions and using the comprehensive personal protective equipment as recommended by professional societies"; it is necessary to mention the numerous national and European sources that dictate these precautionary rules.

We want to thank the reviewer for this suggestion.

The recommendation in question is a standard operation procedure (SOP) distributed among the members of the professional societies, which has not been published. Unfortunately, it can thus not be cited properly. The reference to the recommendations of the professional societies was removed from the two sentences.

4) The stains EvG, Fe, PAS, abPAS, must be preceded by the whole word (Elastica-van-Gieson, Berlinerblau, Periodic acid-Schiff stain, Alcianblau-PAS) before the acronym.

We would like to apologize for the inaccurate description and have made all necessary additions. The full term was added to the acronyms of the stains.

5) "Macroscopic signs of severe and extensive lung damage were found in all patients. In patients 1, 3-5, 9 and 10, the lungs with prominent hyperemia and edema displayed a fragility of the tissue". What does it mean?

We want to thank the reviewer for the comment.

The sentence was intended to express, that the lung tissue – alveoli as well as interstitium – was filled with blood (hyperemia) and fluid (edema) and was at the same time fragile or brittle to pressure. We rewrote the sentence hoping, that the meaning became somewhat clearer.

6) I do not think it is appropriate to keep these descriptive self-satisfaction phrases in the text as the number of cases is still small. "The sample size of our case series can be compared to the larger autopsy case series from literature".

We want to apologize for the phrases in the text and thank the reviewer for bringing this topic to our attention. We rewrote the sentence focusing on the limitation due to the small numbers. We have now deleted the sentence and added:

“The small number of patients is a limitation of the study. However, we have analyzed a large number of organs and tissues and the present study focused on keeping the post-mortem interval as small as possible to avoid bias due to degradation of SARS-CoV-2 virus particles, SARS-CoV-2 RNA and tissue ultrastructure.”

7) "multifocal acute injury of cardiomyocytes which is frequently observed in critically ill patients under catecholamine therapy". The authors must specify the type of cardiac damage by representing whether it is a coagulation myocytolysis or other.

We want to apologize for the inaccurate description. We specified the type of cardiac damage in the text (contraction band necrosis).

8) Discussion paragraphs thirteen and fourteen can be easily removed.

Modification has been made; we omitted the last summarizing paragraph as suggested.

9) Figure 4 is very trivial.

We understand the concerns of the reviewers, but with our article we also want to address scientists who are not pathologists and for them the presentation of a lung damaged by SARS-CoV-2 is very important and impressive. Figure 4 demonstrates macromorphological findings in unfixed organs, which is a complement to the histopathology. We hope that the reviewer agrees with this.

10) Figure 7C-D I do not understand the usefulness of showing a normal myocardium in c and I do not see any alteration of the myocardium in d.

Figure 7D represents a higher magnification of Figure 7C. The patient showed high viral loads in the cardiac samples. The figures emphasize our main result of normal micromorphology or unspecific findings despite high viral loads.

11) The references must also be expanded for the autopsy case reports that have been published and that are not mentioned in this paper.

We have conducted a recent pubmed research using "autopsy and COVID-19". The most important papers have been included in the discussion and quoted accordingly. For this, we have added four case reports and six autopsy studies.

The main message is that COVID-19 is a systemic disease as demonstrated by presence of virus in organs outside the lungs. The description of tissue damage is mostly about lung. Apparently, a correlation viral load – tissue damage could not be established outside lung, while some lesions where present in other organs. The claim that these data allows the study of a correlation between viral load and tissue damage is not fully justified. A few limitations:12) Quantification of viral load insufficiently explained. Materials and methods/Sars-cov2 detection quantification expressed as copies/ml: how did the investigators standardize for amount of tissue in medium

We want to thank the reviewer for the comment and want to apologize for the insufficient description. We have added subsection “SARS-CoV-2 RNA detection*”*

13) No TEM in other organs beside lungs. PCR detects all RNA, not necessarily intact viral particles, so unclear if this is just degraded virus, being transported from lung though the body.

We want to thank the reviewer for the comments. We added a detailed discussion of this issue.

“TEM investigations of the lung samples revealed the presence of morphologically intact virus particles in the tissue which corresponds to the vRNA mapping, which shows the highest viral loads in the lungs. The morphologically intact virus particles were localized in lung fibrocytes.

In agreement with data published by Varga et al. ^49^ viral inclusion bodies were detectable using TEM. Whereas in lung tissues the morphology of viral particles was clearly given (Figure 2), in other tissues like liver, heart, and intestine viral inclusions were not visible with TEM. The loss of structural hallmarks could be due to the post-mortem cell and tissue turnover with reduced integrity of virus particles for TEM-related preparations. It has to be stressed that the molecular detection of the virus depends not on the morphological intact virus particle.”

14) The digestive tract and pancreas not exactly great for post mortem studies, even when the interval between exitus and autopsy is short. Subtle changes may be missed.

Both patients showed high viral loads in the gut and pancreas. The figures emphasize our main result of normal micromorphology or unspecific findings despite high viral loads and demonstrate the very well preserved postmortem histomorphology due to the early postmortem sampling.

15) It is unclear from what area of organs exactly and whether lesional areas were specifically sampled or not. For example, was kidney cortex was properly sampled? So insight in viral tropism and where damage is to be expected is limited.

The samples were taken following routine protocols. In case of the kidney cortex and medulla were sampled. The institute of forensic medicine and the section of surgical pathology are accredited and follow process-oriented quality management.

16) Role of antiviral drugs (potentially interfering with this correlation) is not discussed.None of these limitations is discussed. Both parts of the conclusion that COVID-19 is 1 a systemic disease and 2 that there is no correlation with tissue damage outside the lung are in my view overstretched.

Thank you very much for the comments of the reviewers. We have added the following essential points in the discussion:

“In our study, 4 of 11 patients were treated with a virustatic medication. None of these four patients showed vRNA in the blood, but one patient showed vRNA in the bone marrow (patient 1). The application of virustatic drugs is currently being investigated in many studies. However, it could not yet be shown that a significant effect can be achieved by application (Cao et al., 2020). However, drugs are often used in severe cases in the ICU with controversial recommendations (Meini et al., 2020). Our data could indicate that lopinavir/ritonavir leads to a reduction of viremia. However, our group size is too small for such a statement, as we could not detect vRNA in the blood, even in untreated patients in some cases.”

We would also like to apologize for the correlation statement.

17) The authors present no (plan to study) data on interconnectedness of inflammation, thrombosis and infection/viral load in their Materials and methods. Correlation of IL-6, coagulation markers, presence of virus is unclear from results and figures. E.g. hemophagocytosis in bone marrow observed in one case and is discussed as possibly related to MAS and cytokine storm. Yet, no mention of IL-6. Hence, the presentations of any correlation comes across as a haphazard fishing expedition.

Hemophagocytosis was observed in more than one case but was not the focus of this study. It will be addressed in another study with reference to histiolymphocytosis-related clinical symptoms. The need of further investigation is also mentioned in the text.

18) The conclusion ("Virus may be able to infect different cell types in different organs and tissues but may not be able to replicate in non-respiratory organs/tissues as efficiently as in the respiratory ones") is not based on the findings themselves hence not a conclusion but it may be discussed as a speculative explanation of the findings.

We want to thank the reviewer for the comment and want to apologize for the insufficient description which has been removed.